# Advancing Cholangiocarcinoma Diagnosis: The Role of Liquid Biopsy and CRISPR/Cas Systems in Biomarker Detection

**DOI:** 10.3390/cancers17132155

**Published:** 2025-06-26

**Authors:** Agne Sidabraite, Paula Lucia Mosert, Uzair Ahmed, Stephen Knox Jones, Aiste Gulla

**Affiliations:** 1Faculty of Medicine, Vilnius University, LT-03101 Vilnius, Lithuania; agne.sidabraite@mf.stud.vu.lt (A.S.); paula.mosert@mf.stud.vu.lt (P.L.M.); 2VU LSC-EMBL Partnership for Genome Editing Technologies, Life Sciences Center, Vilnius University, 10257 Vilnius, Lithuania; uzair.ahmed@gmc.vu.lt (U.A.); stephen.jones@gmc.vu.lt (S.K.J.); 3Institute of Clinical Medicine, Faculty of Medicine, Vilnius University, LT-03101 Vilnius, Lithuania

**Keywords:** cholangiocarcinoma (CCA), liquid biopsy (LB), CRISPR/Cas, tumor heterogeneity, early diagnosis, tumor biomarker detection, minimally invasive diagnostics, micro-RNAs (miRNAs)

## Abstract

The diagnosis of cholangiocarcinoma (CCA) poses significant challenges in early and accurate diagnosis due to its variability in etiology and morphology. Existing diagnostic methods often face difficulties in identifying the disease early enough, when surgical treatment could be curative. A poor prognosis and the inaccuracy of current methods emphasize the urgency for a more precise diagnostic tool. Liquid biopsy (LB) has emerged as a promising, less invasive solution for detecting tumor biomarkers from body fluids, such as serum, urine, and bile. Specific biomarkers such as microRNAs, particularly miRNA-21, show a high sensitivity and specificity for CCA. Incorporating CRISPR/Cas systems into liquid biopsy-based diagnostics may offer a rapid and reliable method for detecting tumor-specific biomarkers. Combined, these two innovative approaches could significantly improve the early detection and management of CCA. This paper examines LB and CRISPR/Cas systems as potential improvements for diagnosing cholangiocarcinoma.

## 1. Introduction

Cholangiocarcinoma (CCA) constitutes a heterogeneous group of tumors that arise from cholangiocytes in the biliary tree. CCAs are classified based on their anatomical site of origin and are subdivided into intrahepatic cholangiocarcinoma (iCCA) (tumor lesion developed in the liver) and extrahepatic cholangiocarcinoma (eCCA), which includes perihilar (pCCA) and distal (dCCA) subtypes [1,2].

The prevalence of CCA is increasing worldwide and accounts for approximately 15% of primary liver cancers and about 3% of gastrointestinal malignancies [2]. While Western countries report rates of 2.1 cases per 100,000 people per year, Eastern countries, especially northwest Thailand, face even higher incidence rates. These differences in incidence rates are likely linked to local risk factors and potential genetic predispositions [3,4]. Common risk factors for developing CCA include chronic biliary inflammation, liver cirrhosis, chronic hepatitis B and C infections, diabetes, obesity, and excessive alcohol consumption [2]. Chronic inflammation and cholestasis stand out among these risk factors, likely increasing exposure of cholangiocytes to inflammatory mediators such as TNF-α, IL-6, Wnt, and COX-2. These mediators contribute to the gradual accumulation of mutations in tumor suppressor genes, proto-oncogenes, and DNA mismatch-repair (MMR) genes [5].

CCA is responsible for about 2% of all cancer-related deaths each year [2]. The insidious onset of CCA, its aggressive behavior, and its resistance to chemotherapy contribute to its high mortality rate. The standard chemotherapy regimen for patients with advanced-stage biliary tract cancer (BTC) who are ineligible for surgical resection consists of gemcitabine and cisplatin [6]. Moreover, the 5-year survival rate after resection is typically below 40%, and less than 5% for non-operable cases [7].

Despite significant advances in cancer diagnostics, the majority of CCA cases are still diagnosed at advanced stages, when curative surgical resection is no longer an option [8,9,10,11,12]. Early detection remains a major challenge due to the lack of sensitive, specific, minimally invasive, and widely available diagnostic tools. Hence, there is an urgent need for novel approaches that can enable timely and accurate diagnosis, thereby improving patient outcomes [13]. This review aims to critically evaluate the current limitations of conventional diagnostic modalities for CCA and to explore the potential of integrating liquid biopsy (LB) with CRISPR/Cas-based technologies as a next-generation strategy for rapid, sensitive, and cost-effective CCA detection (Figure 1).

## 2. Materials and Methods

This paper provides a narrative review assessing the potential of combining liquid biopsy technologies with CRISPR/Cas systems to enhance cholangiocarcinoma diagnosis.

The research was conducted in major scientific databases, including PubMed, Web of Science, Google Scholar, and Embase. To ensure up-to-date information and capture the significant developments in the field, articles published between 2018 and 2025 were included. The following key terms and MeSH terms were searched in combinations: “cholangiocarcinoma”, “bile duct neoplasms”, “liquid biopsy”, “biomarkers, tumor”, “microRNAs”, “CRISPR-associated system”, and “precision medicine”. Original articles and publications written in English were selected for analysis. By including relevant studies on early CCA diagnosis, the focus is on analyzing data regarding the detection of tumor biomarkers, specifically microRNAs in body fluids. Additional context literature was sourced from PubMed, BioMed Central, Elsevier, Springer Nature, MDPI, and the Journal of Hepatology. The figures in this review were made with BioRender.com (Toronto, ON, Canada).

## 3. Results

### 3.1. Overview

This narrative review analyzes studies published between 2018 to 2025, sourced from major databases such as PubMed, Web of Science, Google Scholar, and Embase, focusing on the early detection of CCA using LB and CRISPR/Cas technologies. Only original articles published in English were included. The included studies predominantly analyze diagnostic biomarkers, especially microRNAs (e.g., miR-21, miR-16, miR-877), in body fluids like serum, urine, and bile from CCA patients. Patient populations in the included studies vary geographically, focusing on both Western and Southeast Asian cohorts, reflecting on differences in CCA etiology and subtype prevalence, including intrahepatic and extrahepatic subtypes. Methodologies include LB techniques for minimally invasive biomarker detection combined with CRISPR/Cas-based assays, such as the RACE platform. These methodologies enable a sensitive, specific, and multiplexed detection of tumor-derived nucleic acids.

Sample sizes among the included studies vary; however, collectively they emphasize the potential of integrating liquid biopsy with CRISPR/Cas systems to overcome limitations of contemporary serum markers (e.g., CA19-9) and improve early and accurate diagnosis of CCA.

### 3.2. Cholangiocarcinoma: Disease Characteristics and Challenges

CCA is a heterogeneous and aggressive malignancy arising from the epithelial cells of the biliary tract, representing the second most common primary liver cancer after hepatocellular carcinoma [2]. As illustrated in Figure 2, it is anatomically classified into intrahepatic (iCCA), perihilar (pCCA), and distal (dCCA) subtypes with distinct clinical and pathological characteristics. The global incidence of CCA varies widely, with much higher prevalence in Southeast Asian countries such as Thailand and Laos, primarily due to endemic liver fluke infections. In contrast, rising iCCA cases in Western countries [14] may be associated with chronic liver diseases, metabolic syndromes, and lifestyle-related risk factors. The disease predominantly affects individuals over 50 years of age, with a slight male predominance [14].

The pathogenesis of CCA involves environmental and molecular contributors. Chronic biliary inflammation—as seen in primary sclerosing cholangitis—promotes cycles of injury and repair, facilitating malignant transformation. Parasitic infections with *Opisthorchis viverrini* and *Clonorchis sinensis* further contribute to carcinogenesis through the release of metabolites that cause DNA damage [5].

On a molecular level, mutations in key regulatory genes (such as *KRAS*, *TP53*, and *IDH1/2),* along with characteristic gene fusions like *FGFR2,* drive tumorigenesis by dysregulating cellular proliferation and survival pathways [5]. Notably, the prevalence of specific mutations differs by etiology and anatomic subtype: IDH1/2 mutations and FGFR2 fusions are enriched in iCCA tumors from Western patients, whereas TP53, and KRAS, along with associated pathways like SMAD4, are more prevalent in fluke-associated and perihilar tumors [15].

The tumor microenvironment also plays a critical role in CCA. It is characterized by dense desmoplastic stroma, cancer-associated fibroblasts, and immunosuppressive cells such as tumor-associated macrophages, which together promote tumor growth and resistance to therapy [15].

CCA is often clinically silent early on, leading to the diagnosis of most cases only after the disease is at an advanced stage, preventing patients from receiving potentially curative treatments. Less than one-third of cases have single lesions less than 3 cm at diagnosis, almost half show regional lymph node invasion, and a quarter have distant metastases present [16]. The patient’s fragile state and advanced illness increase the risk of bleeding and peritoneal seeding, and the small amount of tissue retrieved may be insufficient for confirmation via cytology or histology [17]. This highlights the urgent need for innovative, less invasive diagnostic strategies—such as the integration of CRISPR/Cas systems with LB technologies—that could facilitate early detection and improve prognosis.

### 3.3. Classification and Working Mechanisms of CRISPR/Cas Systems

CRISPR/Cas (clustered regularly interspaced short palindromic repeats and CRISPR-associated proteins) is an adaptive immune system found in prokaryotes. It allows microorganisms to recognize and defend against viral infections by typically destroying foreign DNA through RNA-guided nuclease activity. This process involves three main steps: adaptation, expression, and interference [18].

Upon viral infection, a bacterium may undergo adaptation: its Cas proteins excise fragments of viral DNA (protospacers) and integrate them into its own genome at the CRISPR locus. This process creates a molecular memory of past infections [18].

During expression, the CRISPR locus is transcribed into a precursor CRISPR RNA (pre-crRNA), which is processed into shorter CRISPR RNAs (crRNAs). Each crRNA contains a single spacer sequence that, after pairing with a trans-activating CRISPR RNA (tracrRNA) in some systems, forms a functional guide RNA (gRNA). This gRNA associates with a Cas protein to direct sequence-specific interference [19].

Finally, interference occurs. The Cas protein identifies a short protospacer-adjacent motif (PAM; typically, 2-5 bp long) that initiates hybridization between the gRNA and the viral protospacer DNA. This gRNA-DNA hybrid licenses the Cas protein to cut the DNA, producing a double-strand break (DSB), which neutralizes the viral genetic material to halt infection [20].

Among several CRISPR/Cas systems [21], type II systems are the most widely studied and recognized. They contain CRISPR-Cas9 nuclease, which cleaves double-stranded DNA and is the most prevalent genome editing tool today [22]. Cas9 may be programmed with a gRNA to target a gene of interest for editing, rather than a viral protospacer. Once Cas9 generates a DSB, cellular mechanisms repair the cut DNA to restore its integrity. These repairs occur through two primary pathways: homology-directed repair (HDR) or non-homologous end joining (NHEJ). Streptococcus pyogenes Cas9 (SpCas9) is the leading variant and recognizes a 3′ NGG PAM sequence. This limits SpCas9′s genome-targeting range, as do potential off-target effects and suboptimal efficiency [21].

To overcome these challenges, researchers have developed a range of CRISPR/Cas systems (Table 1) and combined their components with other enzymes for new activities and reduced toxicity. Base-editing systems enable single-nucleotide modifications without introducing DSBs, while prime-editing systems enable templated insertions or deletions of tens of base-pairs [22]. Beyond its natural function in bacterial defense, CRISPR has been successfully adapted for various molecular biology applications, including genome editing, functional genomics, and molecular diagnostics [20].

Cas12a (Cpf1), a Cas nuclease derived from a Type V CRISPR/Cas system, targets DNA and differs from Cas9 by recognizing a 5′ TTTV PAM and generating sticky ends, which may improve the efficiency of HDR-mediated insertions. In 2018, Chen et al. discovered that some Cas12a nucleases can indiscriminately cleave single-stranded DNA (ssDNA), enabling the detection of viral DNA in patient samples and establishing its potential as a diagnostic tool [19].

Cas12f, formerly known as Cas14, is an exceptionally compact nuclease of the Type V-F CRISPR/Cas system. Initially believed to cleave only single-stranded DNA, recent studies have shown that Cas12f can also recognize T- or C-rich 5′ PAMs and induce double-stranded DNA cleavage, producing staggered 5′ overhangs. Like Cas12a, Cas12f exhibits collateral ssDNA cleavage upon target binding, making it a promising tool for both genome editing and nucleic acid detection [23].

Type VI systems, represented by Cas13, uniquely target single-stranded RNA (ssRNA) rather than DNA, allowing for the editing and modulation of key RNA molecules such as messenger RNAs (mRNAs), microRNAs (miRNAs), and long non-coding RNAs (lncRNAs). This RNA-targeting capability offers great potential for treating cancer and other diseases without making changes to a patient’s genome [24].

**Table 1 cancers-17-02155-t001:** Characteristics of different CRISPR/Cas systems.

CRISPR/Cas	Target Molecule	PAM (PFS) Sequence	Cleavage Domains	Collateral (Trans) Cleavage	Ref.
*Sp*Cas 9	dsDNA	3′ NGG	2	No	[25]
*As*Cas 12a	dsDNA/ssDNA	5′ TTTV	1	Yes	[26]
Cas 12f	dsDNA/ssDNA	Varies by subtypes	1	Yes	[23]
Cas 13d	RNA	Non-G PFS	1	Yes	[27]

### 3.4. Advancements in Liquid Biopsy Technologies for Early Diagnosis of Cholangiocarcinoma

Current serum markers for CCA, including carcinoembryonic antigen (CEA), carbohydrate antigen 19-9 (CA 19-9), and carbohydrate antigen 125 (CA 125), have poor diagnostic accuracy, requiring histopathological confirmation for definitive diagnosis [7]. These markers often exhibit variable sensitivity and specificity, leading to false-negative or false-positive results. Such discrepancies are particularly problematic in early-stage disease, where a small tumor burden may yield low biomarker concentrations in the bloodstream. Furthermore, elevated serum levels of CEA and CA 19-9 are not specific to CCA and can be observed in other malignant and benign conditions, limiting their diagnostic utility [9]. Importantly, these markers do not distinguish different CCA subtypes, complicating accurate differentiation among anatomical variants [9].

Histological variations influence biomarker levels but do not offer clear distinctions among CCA subtypes. As a result, tissue sampling and histopathological examination remain the gold standard, despite their limitations [7]. Tumor tissue collection typically requires risky invasive procedures, which may yield insufficient material for analysis and are often infeasible for tumors in some locations [12]. To overcome these diagnostic challenges, emerging research is increasingly focusing on novel LB biomarkers, which hold promise for improving the accuracy, accessibility, and reducing invasiveness in CCA diagnostics [28].

LB involves analyzing biological fluids such as blood serum, plasma, urine, and bile to detect components derived from the tumor itself or the body’s response to the tumor. Components analyzed from LB samples include circulating tumor cells (CTCs), cell-free DNA (cfDNA), circulating tumor DNA (ctDNA), RNA, and extracellular vesicles (EVs) [29,30]. The analysis of such biomarkers using LB aims to provide less invasive but more accurate CCA diagnoses, particularly for early stages [12].

Sample heterogeneity can lead to misdiagnosis of traditional biopsy specimens, whereas LB samples are more homogenous [29]. Unlike tissue biopsies, LB avoids complex processing steps such as fixation, embedding, or freezing prior to undergoing immunohistochemistry [31]. As a result, chemical preservatives that might interfere with downstream molecular analyses may no longer be required. Therefore, LB can provide fresh tumor-derived components, such as circulating microRNAs.

LB eliminates the risk of invasive biopsy procedures, can be carried out rapidly, and provides accurate genomic, proteomic, and metabolomic information about the tumor [32]. Additionally, it offers the possibility to assess tumor heterogeneity, to identify targeted therapeutic agents based on tumor biology, and to evaluate treatment response and mechanisms of resistance to chemotherapy [12]. Nevertheless, conventional cancer biomarker detection methods, such as PCR and ELISA, typically require large-scale instruments, skilled operators, and laboratory conditions. In contrast, in vitro diagnosis using CRISPR/Cas technology can streamline detection procedures while enhancing sensitivity and specificity [33]. Such advancements in LB technologies could revolutionize the early detection and monitoring of CCA, ultimately improving patient outcomes.

### 3.5. The Role of MicroRNAs and Extracellular Vesicles in Cholangiocarcinoma Diagnosis

Biomarkers are valuable tools for improving cancer diagnosis and monitoring [32]. ctDNA, a product released into the bloodstream from deceased tumor cells, shares its genetic composition with its tumor cell origin. Its short half-life renders it valuable for early cancer detection and real-time monitoring of tumor development, therapeutic response, and outcomes, but can make it harder to capture from freely circulating blood [34].

Alongside ctDNA, EVs hold promise as potential biomarker carriers, including proteins [10] and non-coding RNA (ncRNA) [35]. ncRNA molecules are transcribed but not translated. These are categorized into small ncRNAs (18 to 200 nucleotides) and long ncRNAs (over 200 nucleotides). miRNAs are small ncRNAs (18 to 24 nucleotides) that are transcribed by RNA polymerase II and then processed to become functionally mature. To avoid degradation, miRNAs are encapsulated by vesicles or bound by Argonaut2 (Ago2) components, granting them remarkable stability and resistance during the storage and handling of patient samples [32]. Furthermore, different cancer types could exhibit unique miRNA profiles, which could be leveraged for diagnosing early disease, monitoring disease progression, or assessing treatment response [36,37,38,39].

#### 3.5.1. MicroRNAs

As in other cancer types, miRNAs play key roles across all phases of biliary carcinogenesis, acting either as oncogenes (onco-miRNAs) or tumor suppressors (oncosuppressor-miRNAs). Their pro- and anti-tumorigenic molecular function significantly influences various cancer traits, such as promoting proliferative signaling, evading cell death, and activating invasion and metastasis [40,41]. A meta-analysis of miRNA profiling in CCA found that 70 miRNAs exhibited increased expression levels, while 48 miRNAs displayed decreased expression levels [41].

Specifically, miRNA-21 overexpression in CCA tissue specimens emerges as a potent oncogene biomarker. miRNA-21 exhibits a high sensitivity and specificity in discriminating between CCA and normal tissues (95% sensitivity and 100% specificity) [7,42] and is associated with a poor overall survival rate (HR = 1.88, 95% CI 1.41–2.51) [8]. A single-center study by the Zhenou group involving 1001 plasma samples from patients diagnosed with iCCA revealed significantly elevated plasma miRNA-21 and miRNA-22 levels when compared to controls. Additionally, increased plasma miRNA-21 levels significantly correlated with larger tumor size (*p* = 0.030). These results provide evidence that miRNA-21 overexpression in plasma can serve as a reliable diagnostic marker for iCCA. Moreover, they established a diagnostic three-marker model (plasma-isolated miRNA-21, miRNA-122, and CA19-9) to accurately discriminate iCCA from controls [43]. This highlights how integrating miRNA analysis with traditional markers could enhance diagnostic precision.

Similarly, low serum levels of miRNA-150 are also a sensitive blood biomarker for diagnosing CCA. Although serum miRNA-150 levels are downregulated in CCA patients (diagnostic sensitivity 91.43%, specificity of 80%), its integration with biomarker CA19-9 significantly enhances diagnostic performance (sensitivity 93.33%, specificity 96.88%) [44]. Furthermore, Jiang et al. demonstrated that combining miRNA-21 and miRNA-221 with ultrasound improves diagnostic precision for CCA associated with hepatolithiasis (AUC = 0.911, sensitivity 77.42%, specificity 97.50%) [45].

Serum miRNA-26a has emerged as a noteworthy diagnostic candidate, showing significantly elevated levels in patients with biliary tract cancers (BTC) relative to healthy controls. Its expression correlates with advanced clinical stage, distant metastasis, and poor survival outcomes, making it an independent predictor of both overall and progression-free survival [46]. Studies have demonstrated its high sensitivity and specificity, proposing miRNA-26a as a robust, less invasive biomarker for early BTC detection [8]. However, conflicting studies describe diminished miRNA-26a serum levels in CCA patients compared to those with primary sclerosing cholangitis (PSC) (AUC of 0.780) [47]. Similarly, serum concentrations of miRNA-122, miRNA-192, miRNA-29b, and miRNA-155 were significantly elevated in patients with CCA compared to healthy controls or those with benign conditions such as primary sclerosing cholangitis [48]. Although these preoperative levels lacked prognostic value, a notable postoperative decline in miRNA-122 levels has been strongly linked to favorable outcomes, suggesting benefits for assessing surgical efficacy and monitoring disease progression.

Circulating miRNAs in plasma have also been studied extensively. Studies on miRNA-16 and miRNA-877 indicate their impressive complementary diagnostic value in dCCA (AUC 0.90, sensitivity 79%, specificity 90%), and their ability to distinguish dCCA from pancreatic ductal adenocarcinoma (PDAC) (AUC = 0.88) [49]. Differential expression of miRNA-423-5p, miRNA-93-5p, and miRNA-4532 was observed in CCA patients from regions endemic to *Opisthorchis viverrini* infections; miRNA-4532 was a particularly promising biomarker for distinguishing CCA cases from controls (AUC = 0.8983) [50]. eCCA patients had elevated miRNA-18a and miRNA-532 levels, suggesting diagnostic potential as standalone markers, though larger studies are needed to confirm their clinical utility [51].

In bile samples collected during endoscopic retrograde cholangiopancreatography (ERCP), multiple miRNAs have demonstrated diagnostic relevance. miRNA-1275 and miRNA-6891-5p were significantly upregulated in patients with BTC and pancreatic cancers [20]. miRNA-340 and miRNA-182 differentiated malignant from benign pancreaticobiliary disease, achieving AUC values of 0.79 and 0.772 in independent cohorts [52]. A bile-based miRNA panel comprising miRNA-125b-5p and miRNA-194-5p further demonstrated its ability to differentiate PDAC from CCA with high accuracy (AUC =  0.815) [53].

Despite these promising findings, challenges such as variability in miRNA detection methods, the lack of standardized protocols, and limited validation across diverse cohorts hinder clinical translation [54]. While many of the circulating miRNAs discussed, such as miR-21, have been associated with CCA, they are also elevated in other malignancies such as PDAC and hepatocellular carcinoma (HCC), as well as in non-malignant conditions like inflammation or diabetes [55,56,57,58]. This overlap raises concerns regarding biomarker specificity and may limit their discriminatory power in clinical settings. To address these issues, future research should prioritize the identification and validation of CCA-specific miRNA signatures, ideally in large-scale, multicenter studies. In parallel, harmonizing pre-analytical and analytical workflows and conducting cost-effectiveness evaluations will be crucial to facilitate the integration of miRNA-based tools into routine diagnostic practice.

#### 3.5.2. Extracellular Vesicles

EV-derived miRNA panels present a promising opportunity for CCA diagnosis, offering enhanced diagnostic value when combined with other biomarkers [59]. EVs are categorized based on size and biogenesis: exosomes (typically ~40 to 160 nm) originate from the endosomal pathway, specifically via the fusion of multivesicular bodies with the plasma membrane [60]. Microvesicles (also referred to as large EVs, ~100 to 1000 nm) are directly formed from the plasma membrane. Apoptotic bodies (500 to 5000 nm) are generated during programmed cell death [61]. Unlike ctDNA released from dying cells, exosomes released by living cells provide more precise insight into active tumor cell activity. Furthermore, the lipid bilayer surrounding exosomes protects their cargo from enzymatic degradation, making them more stable than ctDNA in biological fluids [62,63].

The diagnostic potential of exosomal miRNAs for CCA has been extensively investigated: studies on bile-derived exosomes highlight their unique diagnostic potential due to their proximity to the tumor microenvironment. For instance, bile-derived miRNAs have demonstrated high diagnostic accuracy, with miRNA-200c-3p showing an AUC of 0.87 for early CCA detection. The integration of serum CA19-9 further enhances diagnostic performance (AUC = 0.906) [64]. In another study, bile exosomal miRNA-451a and miRNA-3619-3p exhibited high sensitivity and specificity for BTCs, with miRNA-3619-3p also serving as a prognostic marker for poorer outcomes [65]. Functional pathway analysis of bile-derived exosomal miRNAs has revealed enrichment in oncogenic pathways such as RAS signaling, which aligns with tumor biology and underscores their relevance in disease pathogenesis [66].

Comparatively, serum- and blood-derived exosomal biomarkers also show promise for less invasive diagnosis. The miRNA-200 family has superior diagnostic accuracy, with serum-derived exosomal miRNA-200c-3p achieving an AUC of 0.93 [67]. This miRNA family is also associated with tumor stage, offering prognostic value. Furthermore, specific serum-derived exosomal miRNAs, such as miRNA-135b and miRNA-214, have been identified as potential biomarkers for biliary diseases, demonstrating statistically significant upregulation in BTC patients [68].

In summary, while bile- and blood-derived exosomes provide valuable diagnostic insights, bile exosomes often demonstrate superior diagnostic performance due to their proximity to the tumor. These findings suggest a complementary role for the two specimen types: bile-derived exosomal biomarkers may be prioritized for localized diagnostics, while serum-based markers could support broader, less invasive screening strategies. Despite their potential benefits in clinical settings, the low abundance of EVs in biological samples necessitates highly sensitive techniques for their isolation and purification, which currently lack standardization.

These and other circulating miRNAs with potential diagnostic value for CCA are summarized in Table 2.

AUC, area under the receiver operating characteristic curve; SEN, sensitivity; SPE, specificity; CCA, cholangiocarcinoma; EV, extracellular vesicles; BTC, biliary tract cancer; PDAC, pancreatic ductal adenocarcinoma; PSC, primary sclerosing cholangitis.

### 3.6. Role of CRISPR/Cas Systems in Cancer Diagnostics

CRISPR-based diagnostics (CRISPR-Dx) is a molecular cancer diagnostic gaining momentum. Originally developed for genome editing applications such as gene knockout and knock-in using Cas9 proteins, [70,71] CRISPR/Cas systems are now used for nucleic acid detection, enabling rapid, sensitive, and specific identification of target sequences [72,73].

These systems rely on programmable gRNAs that direct a Cas protein (such as Cas9, Cas12, or Cas13) to recognize specific DNA or RNA targets [20]. Upon binding, activated Cas proteins cleave their target (cis-cleavage), which can then trigger collateral cleavage of non-target reporter molecules (as seen with some Cas12 and Cas13 proteins) [73]. These reporter molecules, often tagged with a fluorophore and quencher, produce a fluorescent signal upon cleavage. This provides a simple and visual detection method, licensed by target-specific cleavage. This mechanism underlies highly sensitive and specific detection platforms for circulating tumor biomarkers such as miRNAs and other ncRNAs obtained via LB [19,74].

Compared to the current gold standard for clinical nucleic acid quantification—reverse transcription quantitative PCR (RT-qPCR), CRISPR-Dx platforms offer several advantages. RT-qPCR achieves high sensitivity (down to attomolar concentrations) and single-nucleotide specificity, but is often constrained by complex primer design, especially for short miRNAs, and is hindered by bulky instrumentation and high costs [75]. The use of stem-loop RT primers, necessary for miRNA-specific reverse transcription, may generate background fluorescence or bias, limiting specificity and reproducibility. These challenges severely restrict the use of RT-qPCR in point-of-care or resource-limited settings [76]. In contrast, CRISPR-based systems offer simplified workflows, miniaturized platforms, and faster turnaround times, making them highly suitable for clinical translation [77,78,79].

Numerous platforms demonstrate the diagnostic versatility of CRISPR-Dx. For instance, Chen et al. developed a multiplexed detection platform using Cas12a’s collateral cleavage activity to detect nucleic acids, demonstrating its utility in rapid and sensitive pathogen diagnostics. This approach powers platforms like DETECTR, which exemplifies the transformative potential of CRISPR in clinical diagnostics [19]. Similarly, Gootenberg et al., with a molecular detection platform named SHERLOCK, showcased the versatility of Cas13a when paired with recombinase polymerase amplification, enabling the detection of low-abundance RNA and DNA sequences with high sensitivity [78].

Beyond pathogen detection, CRISPR technology shows significant promise in cancer diagnostics, particularly in the detection of miRNAs—key regulators of gene expression linked to cancer progression. A novel, isothermal amplification platform leveraging CRISPR/Cas9 enables rapid, cost-effective miRNA detection [80]. This marked the first application of the CRISPR/Cas9 system in this context. Furthermore, advanced platforms combining CRISPR/Cas9 with rolling circular amplification (RCA) allowed for the simultaneous detection of multiple miRNAs derived from extracellular vesicles [81]. Despite initial success, Cas9-based multiplexing remains technically challenging due to the need for specific PAMs, signal crosstalk, and absent collateral cleavage activity. As a result, alternative Cas proteins are more frequently employed in diagnostic platforms.

A study utilizing the RCA technique combined with the Cas12a endonuclease successfully detected specific miRNAs within EVs from patients with PDAC at single-digit femtomolar concentrations, achieving single-nucleotide specificity [82]. This multiplexing capability is particularly valuable for early cancer detection, such as CCA, where profiling multiple miRNAs improves diagnostic precision and robustness [83]. Notably, one recent study utilized a CRISPR/Cas13a-based system with machine learning to detect a panel of fecal extracellular vesicle miRNAs (FEVOR) for noninvasive colorectal cancer diagnosis, achieving more than 97% accuracy. This outperforms conventional biomarkers and highlights the diagnostic potential of CRISPR-based miRNA profiling [84].

Isothermal amplification strategies—including loop-mediated isothermal amplification (LAMP) [85], exponential amplification reaction (EXPAR) [86], hybridization chain reaction (HCR) [87], catalyzed hairpin assembly (CHA) [88,89], and rolling circle amplification (RCA) [90]—have been developed for miRNA detection due to their simplicity and low equipment requirements. However, many suffer from limited sensitivity and non-specific amplification, constraining their clinical utility. To overcome these challenges, newer approaches have integrated CRISPR/Cas systems with isothermal amplification in one-pot reactions, enabling enhanced specificity and sensitivity. Chi et al. integrated CRISPR-Cas14a with an isothermal strand displacement amplification (SDA) strategy, demonstrating the ability to detect miRNA-21 in a single test tube within one hour, effectively distinguishing CCA patients from healthy control patients [91]. Other isothermal amplification strategies, such as EXPAR [92], RCA [81,93], LAMP [94], and CHA [95], with assistance of different CRISPR/Cas systems, have been explored for miRNA detection. Despite these advances, most of these methods still involve additional miRNA extraction steps and detect only a single miRNA per assay. Addressing these limitations, Zhang et al. developed a one-pot, one-step CRISPR/Cas13a-based system—termed CRISPR-circuit—that integrates molecular circuitry with CRISPR-mediated cleavage. Operating at attomolar sensitivity within just 15 min, this system eliminates complex workflows, reduces contamination risk, and supports simultaneous analysis of several miRNA biomarkers, representing a significant step forward in point-of-care cancer diagnostics [96].

As the field continues to advance, these innovative technologies hold immense potential to revolutionize cancer diagnostics, providing unprecedented sensitivity and specificity in detecting disease biomarkers. However, ongoing challenges—such as multiplexing, standardization, and clinical validation—must be addressed before widespread clinical implementation. Moreover, there is almost no CRISPR/Cas research specifically focused on molecular profiling of CCA. To address this gap, we propose an exemplary research protocol that leverages CRISPR-based miRNA detection to enhance early diagnostic accuracy for this aggressive and often late-diagnosed cancer.

### 3.7. Challenges and Future Directions of the CRISPR/Cas-Guided CCA Diagnosis

Despite the promising developments in CRISPR-based technologies for LB of CCA, several knowledge gaps remain. Studies demonstrate the potential of CRISPR systems in in vitro laboratory settings but require clinical validation in diverse patient populations. Their integration into routine clinical practice poses logistical challenges, including standardization and regulatory approval. CRISPR detection mainly depends on sensitive fluorescence techniques, but requires specialized excitation and detection equipment, making it challenging to utilize in point-of-care settings [97]. While CRISPR-based systems can detect low-abundance targets, they are still underdeveloped in terms of sensitivity and specificity. This may be caused by cross-reactivity, which results in false-positive or false-negative results [97].

Future research should focus on the longitudinal assessment of biomarkers in CCA patients through LB, enabling disease progression and treatment monitoring. Moreover, studies investigating the interplay between biomarkers, such as miRNAs and other molecular pathways in CCA, could provide deeper insights into the tumor microenvironment, potentially revealing novel therapeutic targets. Standardization protocols should be employed to ensure consistent results among laboratories and departments before CRISPR systems can be integrated into routine diagnostics of CCA [97].

## 4. CRISPR/Cas9-RACE for miRNA-Based Detection of Distal Cholangiocarcinoma: A Diagnostic Protocol

### 4.1. Objectives

One of the major challenges in clinical oncology is the differentiation between dCCA and other anatomically and clinically similar cancers, particularly PDAC, which can lead to misdiagnosis and suboptimal treatment [98]. MiRNAs have emerged as promising, less invasive biomarkers to improve diagnostic precision. A previously identified two-miRNA panel—miRNA-16 (downregulated) and miRNA-877 (upregulated)—showed strong discriminatory ability between dCCA, PDAC, and benign conditions (AUC = 0.90 vs. benign; 0.88 vs. PDAC) [49]. However, conventional RT-qPCR methods are labor-intensive and may lack sensitivity for low-abundance miRNAs.

This chapter focuses on how liquid biopsy and CRISPR technologies could be optimized not only for early detection but also for differential diagnosis between dCCA and related malignancies, such as PDAC. In this context, we propose evaluating the diagnostic potential of a Rolling Circle Amplification-assisted CRISPR/Cas9 cleavage (RACE) platform for the ultrasensitive detection of EV-derived miRNAs in plasma. Specifically, the study aims to determine whether the RACE assay can enhance diagnostic accuracy in distinguishing dCCA from PDAC and benign biliary conditions, while also assessing its performance relative to RT-qPCR and its feasibility for routine clinical use.

### 4.2. Methods

#### 4.2.1. Study Population

A prospective cohort of adult (≥18 years) patients will be enrolled into four groups:(1)dCCA group: Patients with histologically confirmed dCCA or diagnosis based on imaging and clinical data when biopsy is not feasible;(2)PDAC group: Patients with pancreatic ductal adenocarcinoma, matched to dCCA cases by age, sex, and, when possible, disease stage;(3)Benign Biliary group: Patients with non-malignant causes of distal bile duct obstruction (e.g., stones, primary sclerosing cholangitis, autoimmune cholangiopathy, chronic pancreatitis);(4)Healthy Controls: Age- and sex-matched individuals without cancer or significant liver/biliary disease, used to define baseline biomarker levels.

#### 4.2.2. Sample Processing and EV Isolation

Blood samples (5–10 mL) will be collected in EDTA tubes, immediately centrifuged at 2000× *g* for 15 min at 4 °C to separate plasma. Extracellular vesicles (EVs) will be isolated using commercial isolation kits optimized for biofluids (e.g., Qiagen miRNeasy Serum/Plasma or Norgen Biotek kits), following the manufacturer’s instructions precisely. RNA concentration will be quantified using a NanoDrop spectrophotometer. Samples yielding RNA in the picogram range will be stored at −20 °C for immediate use or at −80 °C for long-term storage.

#### 4.2.3. miRNA Detection with RACE

The assay will follow the method described by Wang et al. (2020 [81]), adapted to detect miRNA-16 and miRNA-877, and involves two key steps (Figure 3):

1. Padlock Probe Ligation and Rolling Circle Amplification (RCA): A padlock probe, complementary to the target miRNA’s 5′ and 3′ ends and containing a PAM site for Cas9, is ligated using high-fidelity ligase. The circularized probe serves as a template for RCA using phi29 polymerase at 37 °C, producing long single-stranded DNA with repeated target sequences.

Mix isolated miRNA with padlock (designed specifically for miRNA-16 and miRNA-877) and high-fidelity DNA ligase in ligation buffer;Incubate the mixture at 37 °C for 30 min to ensure complete circularization;Add phi29 polymerase, dNTPs, and RCA buffer to the ligated mixture;Incubate at 37 °C for 60 min to generate long, single-stranded DNA with repeated miRNA-complementary sequences;Heat-inactivate phi29 polymerase at 65 °C for 10 min.

2. CRISPR/Cas9 Detection: A pre-assembled Cas9/sgRNA complex targets a specific sequence in the RCA product. Upon binding, Cas9 cleaves the DNA, triggering fluorescence via a TaqMan-style reporter probe labeled with a fluorophore (e.g., FAM) and quencher (e.g., BHQ1). Cleavage separates the fluorophore from the quencher, restoring fluorescence. One miRNA molecule can thus lead to multiple detection events, amplifying the signal. Fluorescence intensity will be measured in real time using a plate reader or RT-qPCR system in detection mode. Calibration curves will be generated using synthetic miRNA standards.

Prepare Cas9/sgRNA complexes: incubate purified Cas9 protein with guide RNA (sgRNA) targeting PAM sequences within RCA products at 25 °C for 10 min;Mix Cas9/sgRNA complexes with RCA products and a TaqMan-style fluorescent reporter probe (FAM/BHQ1 labeled);Incubate at 37 °C for 30 min at dark conditions; Cas9 cleavage separates fluorophore from quencher, resulting in fluorescence emission;Measure fluorescence intensity using a fluorescence spectrophotometer;Generate calibration curves with known synthetic miRNA standards to quantify miRNA levels accurately.

#### 4.2.4. Validation with RT-qPCR

Parallel RT-qPCR validation will be performed on identical RNA samples using TaqMan stem-loop RT primers specific for miRNA-16 and miRNA-877, following the manufacturer’s guidelines. Ct values will be directly compared to RACE-derived fluorescence intensities to validate assay reliability.

#### 4.2.5. Statistical Analysis

Differences in miRNA expression between groups will be analyzed using *t*-tests or Mann–Whitney U tests. ROC curves will be constructed to determine diagnostic performance (AUC, sensitivity, specificity). A logistic regression model combining multiple miRNAs will be developed to classify dCCA.

#### 4.2.6. Expected Outcomes and Implications

We expect that the RACE platform will demonstrate high diagnostic accuracy for dCCA. The ultrasensitive nature of RACE may enable detection of lower-level miRNA expression, potentially identifying disease at earlier stages. Furthermore, the assay’s single-nucleotide specificity and rapid turnaround time (~37 °C, no thermocycling) support its suitability for point-of-care diagnostics.

Successful implementation of this assay could transform the diagnostic approach to pancreaticobiliary malignancies, providing a minimally invasive blood test to differentiate dCCA from PDAC and benign conditions, thus improving therapeutic decision-making and patient outcomes.

## 5. Discussion

CCA presents a significant challenge in oncology due to its heterogeneous nature, diverse etiologies, and complex molecular alterations [2]. This complexity translates into difficulties in accurate diagnosis, effective management, and ultimately, poor prognoses for affected individuals [7].

Traditional diagnostic tools—biopsy, imaging, and serum biomarkers (e.g., CA19-9, CA125, and CEA) are considered the gold standard in CCA diagnostics, since they show excellent accuracy in the monitoring of disease severity, progression, and response to treatment [7]. However, they struggle to achieve high sensitivity and specificity at low serum concentrations of biomarkers. This means that the existing diagnostic modalities for CCA often fail to detect the disease at its early stages, when curative interventions are most feasible [9]. Current methods also cannot differentiate between malignant and benign diseases, as well as the anatomical origin of the tumor, and the high morphological heterogeneity within the disease [12]. Therefore, they are inadequate for early detection, emphasizing the urgent need for novel approaches.

LB has emerged as a promising method that safely bypasses tissue biopsy sample heterogeneity and preservative contamination by utilizing fresh body fluid samples [12]. Not only can it be carried out more safely and quickly than a conventional invasive biopsy, but it also yields accurate genomic, proteomic, and metabolomic information about the tumor [32]. LB offers a less invasive alternative, with miRNAs emerging as potential biomarkers, amid a myriad of other biomarkers under investigation [40]. Notably, miRNA-21 overexpression in CCA tissue specimens has demonstrated promising diagnostic utility, exhibiting impressive sensitivity and specificity in discriminating between cancerous and normal tissues [42]. Still, conventional miRNA detection (e.g., RT-qPCR) poses challenges: low abundance, short sequence length, and variability in amplification protocols [75,76]. Current less invasive detection methods, including LB, require enhancements in sensitivity, specificity, and cost-effectiveness [29].

CRISPR-based diagnostics (CRISPR-Dx) overcome these hurdles by leveraging programmable nucleases like Cas12 and Cas13 for precise, amplification-free detection of miRNAs [19,74]. CRISPR/Cas systems offer the ability to target specific nucleic acid sequences with unparalleled accuracy, making them well-suited candidates for detecting miRNAs and other biomarkers linked to CCA [80,82]. Paired with isothermal amplification methods such as RCA, these platforms achieve attomolar sensitivity and real-time signal output [81].

The proposed RACE platform, targeting miR-16 and miR-877, we hope, could stand out as a scalable, highly specific detection tool. Its capacity for multiplexed analysis and integration with simple detection systems (e.g., fluorometric plate readers) opens new opportunities for point-of-care use, especially in resource-limited settings.

Further research and development in this area could significantly advance CCA diagnosis and prognosis, potentially revolutionizing patient care and outcomes. Although CRISPR/Cas systems have primarily been employed for genome editing applications, their diagnostic potential—particularly in targeting non-coding RNA regions such as miRNAs—remains underexplored and requires extensive validation in clinical trials [97]. However, cross-reactivity with non-target sequences can result in false positives and deviations in sensitivity, complicating result interpretation [97]. Challenges in technology standardization and protocol application must be addressed to ensure safe and effective integration of liquid biopsy with CRISPR/Cas systems into clinical practice. Establishing standardized protocols across laboratories and clinical departments, advancing technological capabilities, and providing specialized training for healthcare professionals will be essential to fully harness the potential of CRISPR/Cas systems in advancing CCA diagnostics and improving patient outcomes [97].

## 6. Conclusions

In conclusion, the integration of liquid biopsy combined with CRISPR/Cas systems represents a promising advancement in the early diagnosis of cholangiocarcinoma. In terms of patient safety, liquid biopsy greatly reduces health risks and overall harm to the patient, which could potentially arise during the process of obtaining tissue biopsies. These methods could enable rapid and cost-efficient biomarker detection, streamline the diagnostic process, and revolutionize patient management by enabling timely interventions. Despite the advancements, strategies combining both methods are still under development and require validation. Comprehensive clinical testing of this diagnostic approach paves the way for its successful implementation into clinical practice. Ultimately, these innovative strategies could have the potential to improve patient outcomes and deepen our understanding of the complexity of cholangiocarcinoma.

## Figures and Tables

**Figure 1 cancers-17-02155-f001:**
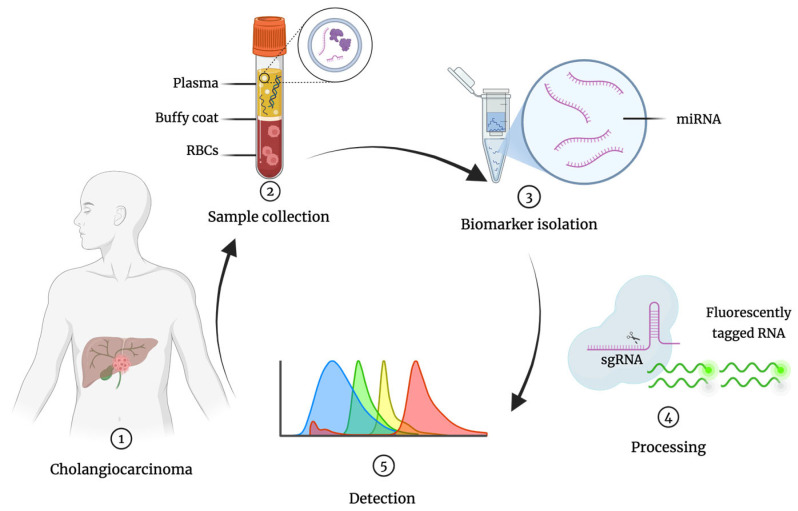
General overview of miRNA detection workflow: 1. CCA patient; 2. Liquid biopsy collection from the patient (e.g., blood) and sample treatment; 3. miRNA isolation; 4. Sample processing; 5. Detection of miRNA (colored peaks represent different target miRNAs). RBCs, red blood cells; miRNA, microRNA; sgRNA, single guide RNA.

**Figure 2 cancers-17-02155-f002:**
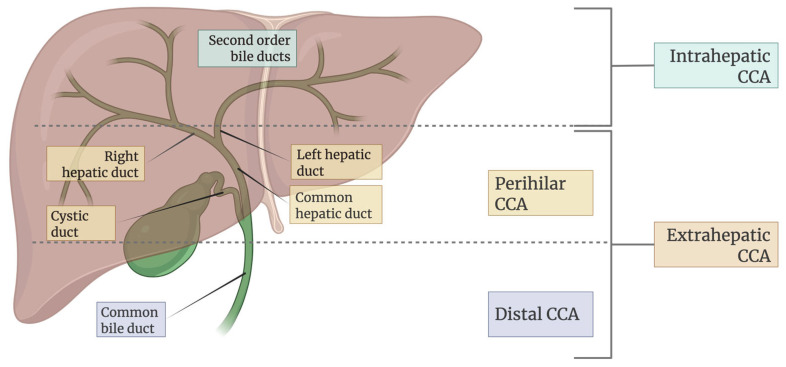
Classification of CCA based on anatomical location in the biliary tree.

**Figure 3 cancers-17-02155-f003:**
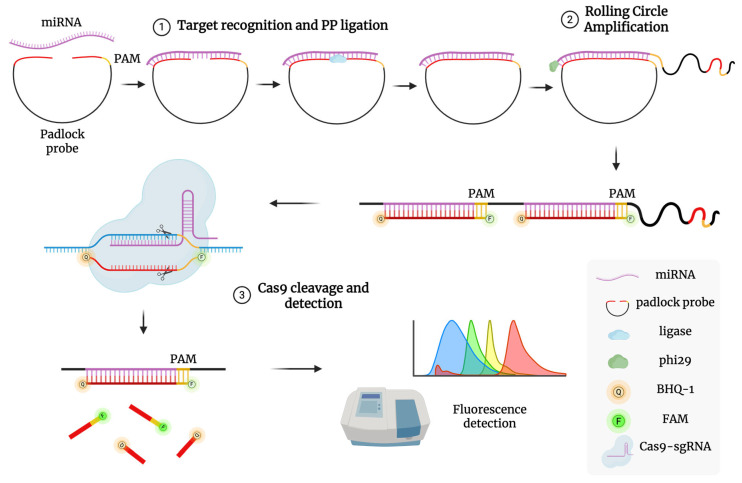
Schematic illustration of RACE reaction mechanism. (1) Target recognition and PP ligation: A PP with a PAM domain binds to the target miRNA, allowing its two ends to be joined into a circular structure by HiFi Taq DNA ligase. (2) Rolling circle amplification: The TaqMan probe hybridizes with the replicated targets to form a long dsDNA structure after RCA replication. (3) Cas9 cleavage and detection: The sgRNA/Cas9 complex recognizes the PAM domain in the dsDNA assembly and cleaves the hybridized duplex, separating fluorophore from quencher and producing a detectable fluorescence signal.

**Table 2 cancers-17-02155-t002:** Circulating miRNAs with potential diagnostic value for CCA.

Detectable Location	miRNA	Levels in CCA	Comparison	AUC	SEN%	SPE%	Ref.
**Tissue**	miRNA-21	Up	CCA (N = 18) vs. healthy controls (N = 12)	0.995	95.00	100.00	[42]
**Plasma**	miRNA-21	Up	Hepatolithiasis-CCA (N = 31) vs. hepatolithiasis (N = 40)	0.610	96.77	30.00	[45]
	miRNA-21 and miRNA-22	Up	iCCA (N = 359) vs. control (N = 642)	0.790	63.90	84.70	[43]
	miRNA-221	Up	Hepatolithiasis-CCA (N = 31) vs. hepatolithiasis (N = 40)	0.767	54.84	95.00	[45]
	miRNA-21, miRNA-221 ^a^	Up	Hepatolithiasis-CCA (N = 31) vs. hepatolithiasis (N = 40)	0.911	77.42	97.50	[45]
	miRNA-16 and miRNA-877	Up	dCCA (N = 24) vs. benign (N = 20)	0.900	79.00	90.00	[49]
		Up	dCCA (N = 24) vs. PDAC (N = 24)	0.880	70.83	90.00	[49]
	miRNA-423-5p, miRNA-93-5p, and miRNA-4532	Up	CCA (N = 30) vs. healthy controls (N = 30) vs. *Opisthorchis viverrini* (N = 30)	-	85.71	76.92	[50]
**Serum**	miRNA-150-5p	Down	CCA (N = 35) vs. healthy controls (N = 35)	-	91.43	80.00	[44]
	miRNA-26a	Up	CCA (N = 66) vs. healthy controls (N = 66)	0.899	84.80	81.80	[46]
		Down	CCA (N = 30) vs. PSC (N = 30)	0.780	52.00	88.00	[47]
	miRNA-122, miRNA-192, miRNA-29b, and miRNA-155 ^b^	Up	CCA (N = 94) vs. healthy controls (N = 40)	-	98.30	100.00	[48]
	miRNA-18a	Up	Extrahepatic CCA (N = 27) vs. healthy controls (N = 13)	0.360	51.80	84.60	[51]
	miRNA-532	Up	Extrahepatic CCA (N = 27) vs. healthy controls (N = 13)	0.350	66.60	69.20	[51]
**Bile**	miRNA-1275	Up	BTC (n = 38) vs. control (n = 35)	0.630	52.60	80.00	[20]
	miRNA-340 and miRNA-182	Up	Cholangiocarcinoma (N = 14) vs. benign disease (N = 37)	0.790	64.60	82.10	[52]
	miRNA-125b-5p and miRNA-194-5p	Up	PDAC (N = 28) vs. CCA (6)	0.815	-	-	[53]
**Bile EV miRNAs**	miRNA-200c-3p	Up	CCA (N = 50) vs. biliary stone (N = 50)	0.870	83.30	86.70	[64]
	miRNA-451a and miRNA-3619-3p	Up	BTC (N = 45) vs. noncancer control samples (N = 43)	0.819	73.50	88.20	[65]
**Serum EV miRNA**	miRNA-200c-3p	Up	CCA (N = 36) vs. healthy controls (N = 12)	0.930	-	-	[67]
**Plasma EV miRNA**	miRNA-194–5p	Down	CCA (N = 15) vs. healthy controls (N = 15) vs. *Opisthorchis viverrini* (N = 15)	-	-	-	[69] ^c^
	miRNA-203a-3p, miRNA-192–5p,	Up		-	-	-	

^a^ When combined with ultrasound. ^b^ At least 2 miRNAs were seropositive. ^c^ Quantitative diagnostic performance metrics were not reported in this study. Biomarker relevance was based on sequencing counts and fold change analysis.

## Data Availability

No new data were created or analyzed in this study. Data sharing is not applicable to this article.

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
