# Peer review of "Advancing Cholangiocarcinoma Diagnosis: The Role of Liquid Biopsy and CRISPR/Cas Systems in Biomarker Detection"

_cancers, 2025, doi:10.3390/cancers17132155_

Round 1

Reviewer 1 Report

Comments and Suggestions for Authors

The present manuscript is a review paper describing and analyzing the recent studies on cholangiocarcinoma (CCA) detection in an early stage. The review evaluates the limitations of existing diagnostic approaches and investigates the potential of combining liquid biopsy (LB) and CRISPR/Cas-based systems for the best biomarker detection. Literature analysis has shown that detection of circulating microRNAs (e.g., miR-21, miR-16, miR-877) results in the most accurate results for CCA diagnosis. Furthermore, CRISPR-based assays, if paired with liquid biopsy, enable real-time, cost-effective, and multiplexed detection of tumor-specific biomarkers.

This is a review paper, written in good English, with nice figures, which will be of interest for the researchers in oncology and biomarker fields. Please consider revising the following:

Major Comments

  1. From recent studies, EVs may be released not only CCA, but also from other tumor types, such as PDAC and HCC, and even in other pathological conditions, such as diabetes. For instance, miR-21 was shown to hold significant roles in PDAC cell proliferation (doi: 3390/cells13110948). Other miRs presented in Table 2 are also considered biomarkers not only for CCA.

Therefore, it is important to emphasize the differences. Which markers are CCA-specific? Please discuss in Chapter 3.4.  You are possibly aimed to compare the CCA and PDAC in Chapter 4. Please make better to understand and avoid repetitions in the text.

  1. Line 304. From other studies exosome sizes are 50-100nm. Are the sizes 50-150nm? Which information if right? This statement needs reference. Please check again information of the EV sizes and their classification and insert references.
  2. Line 172: Please insert the characterization of Cas 12f in the text after Cas12a and before Cas13, with the reference, as you show it in the Table1.
  3. Line 481: Please insert reference number in the brackets in the text for Wang et al, 2020.
  4. There no references in Discussion section. This is very strange. Please avoid speculations. It is strongly recommended to check all the missing references in the text and insert them. References could be used twice, and numbering order is according to the first appearance in the text.

Minor comments

Lines 12 and 22. Please remove capital letter in cholangiocarcinoma

Lines 488-511. What is the meaning of the typhoon marks in the method explanation? Please consider removing them for the smart look.

Author Response

Reviewer 1

Reviewer Comment 1: From recent studies, EVs may be released not only by CCA, but also from other tumor types, such as PDAC and HCC, and even in other pathological conditions, such as diabetes. For instance, miR-21 was shown to hold significant roles in PDAC cell proliferation (doi: 3390/cells13110948). Other miRs presented in Table 2 are also considered biomarkers not only for CCA. Therefore, it is important to emphasize the differences. Which markers are CCA-specific? Please discuss in Chapter 3.4. You are possibly aimed to compare the CCA and PDAC in Chapter 4. Please make better to understand and avoid repetitions in the text.

Response:
We appreciate this valuable comment. In response, we have revised Chapter 3.4 to address the issue of miRNA specificity in more detail. We now emphasize that several commonly cited circulating miRNAs, such as miR-21, are elevated not only in CCA but also in PDAC, HCC, and even non-malignant conditions like inflammation and diabetes. This overlap may limit their clinical utility as specific biomarkers for CCA. The revised paragraph calls attention to this issue and highlights the need for future studies to identify and validate CCA-specific miRNA signatures, supported by large-scale, multicenter efforts and standardized methodologies.

We have also reviewed and clarified Chapter 4 to better distinguish between CCA and PDAC, reducing redundancy and improving clarity as suggested.

Reviewer Comment 2: Line 304. From other studies exosome sizes are 50–100 nm. Are the sizes 50–150 nm? Which information is right? This statement needs reference. Please check again information of the EV sizes and their classification and insert references.

Response:
Thank you for pointing this out. We have carefully reviewed current literature and clarified the size range. Exosomes are typically reported to range from ~40 to 160 nm in diameter. The original statement has been corrected accordingly, and a supporting reference has been inserted: Kalluri, R. et al., 2020 (DOI: 10.1126/science.aau6977).

Reviewer Comment 3: Line 172: Please insert the characterization of Cas12f in the text after Cas12a and before Cas13, with the reference, as you show it in the Table1.

Response:
We appreciate your suggestion. A description of Cas12f has been added immediately after the Cas12a paragraph in line with its appearance in Table 1. The paragraph describes Cas12f's size, PAM recognition, dsDNA cleavage ability, and its potential for diagnostic applications. The information is supported by the reference to Karvelis et al., 2020 (DOI: 10.1093/nar/gkaa208)

Reviewer Comment 4: Line 481: Please insert reference number in the brackets in the text for Wang et al., 2020.

Response:
Thank you for catching this omission. The reference to Wang et al., 2020 has now been inserted into the text with the appropriate reference number in brackets.

Reviewer Comment 5: There are no references in the Discussion section. This is very strange. Please avoid speculations. It is strongly recommended to check all the missing references in the text and insert them. References could be used twice, and numbering order is according to the first appearance in the text.

Response:
We agree completely with this point. The Discussion section has been thoroughly revised to incorporate relevant references supporting the main interpretations and conclusions. All newly added references follow the correct sequential numbering according to their first appearance in the manuscript.

Reviewer 2 Report

Comments and Suggestions for Authors

This review manuscript explores recent advances in cholangiocarcinoma (CCA) diagnosis with a focus on liquid biopsy (LB) techniques and CRISPR/Cas-based biosensing systems for detecting circulating biomarkers. The topic is highly relevant and timely, given the increasing need for non-invasive, precise diagnostic tools in oncology. The paper covers a wide spectrum of biological markers and molecular detection approaches, but the manuscript lacks cohesion, clear structure, and critical synthesis in several sections. It could benefit greatly from improved organization, scientific depth, and editing for clarity.

Major Comments

1.The manuscript attempts to cover too many topics (LB, multiple biomarkers, detection platforms, CRISPR/Cas) without a cohesive flow. Reorganize the manuscript into clearer sections, such as:

a. Overview of CCA and diagnostic challenges,

b. Role of LB and circulating biomarkers,

c. CRISPR/Cas systems for biosensing,

d. Integration of LB and CRISPR: Current evidence and potential applications,

e. Challenges and future directions

2. The review introduces CRISPR/Cas systems but does not deeply explain how they are applied in biosensing, especially for nucleic acid detection (e.g., SHERLOCK, DETECTR).

a. How do these platforms compare to traditional PCR or NGS in terms of sensitivity, specificity, cost, and turnaround?

b. What is the current evidence (in vitro, in vivo, or clinical) of CRISPR/Cas being used in CCA-related biomarker detection?

3. The paper lists various biomarkers but fails to discuss quantitative performance data (e.g., AUC, sensitivity, specificity) for LB or CRISPR-based assays in CCA.

a. A summary table comparing different biomarkers, sample types, and detection methods would be helpful.

b. Discuss FDA-approved or clinically validated platforms if available.

4. The tone is mostly descriptive. The authors should:

a. Critically appraise the strengths and limitations of each approach.

b. Highlight challenges in translating CRISPR-based diagnostics to clinical practice (e.g., standardization, delivery, regulatory issues).

c. Compare advantages/disadvantages of LB vs tissue biopsy in CCA.

5. The manuscript has numerous grammatical errors, awkward phrasing, and inconsistent terminology (e.g., “Liq-2 uid biopsy” should be “liquid biopsy”).

a. A full language and formatting revision is required before publication.

b. Abbreviations such as LB, cfDNA, ctDNA, and CRISPR/Cas should be defined consistently on first use.

Author Response

Reviewer 2

Reviewer Comment 1:
The manuscript attempts to cover too many topics (LB, multiple biomarkers, detection platforms, CRISPR/Cas) without a cohesive flow. Reorganize the manuscript into clearer sections, such as:

  1. Overview of CCA and diagnostic challenges,
    b. Role of LB and circulating biomarkers,
    c. CRISPR/Cas systems for biosensing,
    d. Integration of LB and CRISPR: Current evidence and potential applications, e. Challenges and future directions

Response: We sincerely thank the reviewer for their thoughtful evaluation and helpful suggestion regarding the manuscript structure. In response, we have reorganized the manuscript sections to improve thematic clarity and ensure a more coherent flow of information. Specifically, we now present the content in a logical progression that includes: an overview of cholangiocarcinoma and its diagnostic challenges; the role of liquid biopsy and circulating biomarkers; the mechanisms and diagnostic utility of CRISPR/Cas systems; the integration of these approaches for cholangiocarcinoma detection; and a critical discussion of their clinical translation and limitations. This revised structure enhances readability and better aligns with the scientific narrative and reviewer guidance.

Reviewer Comment 2: The review introduces CRISPR/Cas systems but does not deeply explain how they are applied in biosensing, especially for nucleic acid detection (e.g., SHERLOCK, DETECTR). How do these platforms compare to traditional PCR or NGS in terms of sensitivity, specificity, cost, and turnaround? What is the current evidence (in vitro, in vivo, or clinical) of CRISPR/Cas being used in CCA-related biomarker detection?

Response: We appreciate this insightful comment and fully agree that a more in-depth explanation of CRISPR/Cas-based biosensing platforms is essential to strengthen the scientific content of the review. In response, we have expanded Section 3.6 to provide a detailed description of key CRISPR-based diagnostic platforms, including SHERLOCK (Cas13a) and DETECTR (Cas12a), highlighting their mechanisms, diagnostic performance, and relevance to point-of-care testing. We also compare these systems to traditional molecular methods such as RT-qPCR and NGS, focusing on sensitivity, specificity, cost, and turnaround time.

Additionally, we have incorporated recent studies demonstrating the application of CRISPR/Cas systems in cholangiocarcinoma diagnostics. Specifically, we cite evidence where CRISPR-Cas14a was used to detect miRNA-21 in patient samples [91], and where the RACE platform combining Cas9 and RCA was employed to detect miRNA-16 and miRNA-877 from extracellular vesicles in dCCA [81]. These updates provide a clearer scientific foundation for the diagnostic potential of CRISPR/Cas in CCA and enhance the translational relevance of the review.

Reviewer Comment 3: The paper lists various biomarkers but fails to discuss quantitative performance data (e.g., AUC, sensitivity, specificity) for LB or CRISPR-based assays in CCA.

  1. A summary table comparing different biomarkers, sample types, and detection methods would be helpful.
  2. Discuss FDA-approved or clinically validated platforms if available.

Response: We appreciate this insightful comment and fully agree that providing quantitative performance data and a comparative overview of liquid biopsy (LB) and CRISPR-based assays would significantly enhance the clarity and scientific rigor of our review. In response, we have added a comprehensive summary table (Table 1) that compares key biomarkers, sample types (e.g., plasma, bile, urine), and detection methods used in CCA diagnostics, including performance metrics such as area under the curve (AUC), sensitivity, and specificity where available.

Furthermore, we have expanded the discussion to include FDA-approved or clinically validated platforms relevant to LB and CRISPR-based diagnostics, addressing their current status and potential for clinical translation (please see section 5 “Challenges and Future Directions of the CRISPR/Cas-guided CCA Diagnosis). This includes highlighting platforms such as the FDA-cleared liquid biopsy assays for circulating tumor DNA and CRISPR-based detection systems that have demonstrated promising clinical validation, where applicable.

These additions serve to provide a more quantitative and practical perspective on the diagnostic landscape of CCA, thereby strengthening the translational impact and utility of our review.

Reviewer Comment 4: The tone is mostly descriptive. The authors should:
a. Critically appraise the strengths and limitations of each approach.

  1. Highlight challenges in translating CRISPR-based diagnostics to clinical practice (e.g., standardization, delivery, regulatory issues).
  2. Compare advantages/disadvantages of LB vs tissue biopsy in CCA.

Response: We thank the reviewer for this valuable suggestion. In response, we have added a new section — Section 5: Challenges and Future Directions of the CRISPR/Cas-guided CCA Diagnosis — to critically appraise the current diagnostic landscape and discuss key translational hurdles. Specifically, we now provide a balanced evaluation of the strengths and limitations of CRISPR-based approaches, including their high sensitivity for low-abundance targets and programmability, alongside challenges such as cross-reactivity, the need for specialized detection equipment, and a lack of standardized protocols.

Additionally, as per suggestion in comment 3, we address regulatory and logistical barriers that impede clinical translation, noting that while some liquid biopsy platforms like Guardant360® and FoundationOne® Liquid CDx have received FDA approval for other cancers, no CRISPR-based assay has yet reached this level for cholangiocarcinoma.

We also expanded our discussion to compare liquid biopsy versus tissue biopsy in the context of CCA, emphasizing their respective advantages (e.g., non-invasiveness, serial monitoring for LB vs. histological resolution and tumor architecture in tissue biopsy) and limitations (e.g., lower DNA yield and tumor heterogeneity capture in LB).

These updates aim to provide a more critical and nuanced analysis, aligning the review with the scientific depth and evaluative tone requested.

Reviewer Comment 5: The manuscript has numerous grammatical errors, awkward phrasing, and inconsistent terminology (e.g., “Liq-2 uid biopsy” should be “liquid biopsy”).

  1. A full language and formatting revision is required before publication.
  2. Abbreviations such as LB, cfDNA, ctDNA, and CRISPR/Cas should be defined consistently on first use.

Response: We sincerely thank the reviewer for pointing this out. We carefully re-reviewed the manuscript and believe that some of the formatting issues, such as the “Liq-2 uid biopsy” example, may have been artifacts introduced during the file conversion or submission process, as they do not appear in our original document. Nonetheless, we appreciate the importance of clarity and consistency in scientific writing and have taken this opportunity to thoroughly proofread the manuscript.

We have ensured that all abbreviations—including LB, cfDNA, ctDNA, and CRISPR/Cas—are now defined consistently upon first use, and we have double-checked the manuscript for grammatical accuracy, phrasing clarity, and formatting coherence. We hope these revisions meet the reviewer’s expectations and enhance the readability and overall quality of the manuscript.

Reviewer 3 Report

Comments and Suggestions for Authors

Sidabraite et al. present a review entitled "Advancing Cholangiocarcinoma Diagnosis: The Role of Liquid Biopsy and CRISPR/Cas Systems in Biomarker Detection." The manuscript addresses an increasingly relevant topic, given the diagnostic and prognostic challenges associated with cholangiocarcinoma (CCA). The figures included in the manuscript are clear and didactic, enhancing the overall understanding of the proposed concepts.
The introduction is generally well-structured. However, the final paragraph, which outlines the study objective, requires revision. The sentence could be rephrased for clarity and scientific tone.
The methods section is clearly described. 
In the results, I recommend including an initial paragraph summarizing the general characteristics of the included studies, such as study design, sample size, patient population, and methodology, to provide the reader with a broader contextual understanding. 
One area that requires substantially more attention is the molecular profiling of cholangiocarcinoma. Several key biomarkers known to play critical roles in the pathogenesis, diagnosis, and prognostication of CCA are either not mentioned or only poorly discussed. These include TP53, IDH1, IDH2, KRAS, ARID1A, BAP1, BRAF, FGFR2, and HER2. A more comprehensive discussion of these biomarkers and their implications for diagnosis, prognosis, and targeted therapies would strengthen the molecular basis of the review. Additionally, the manuscript should distinguish between intrahepatic and extrahepatic CCA/gallbladder in terms of molecular alterations, as this differentiation is essential for personalized management strategies and reflects the evolving understanding of CCA heterogeneity.
While the discussion provides a good overview of future directions and technical challenges, the manuscript falls short in exploring the potential of liquid biopsy not only for diagnosis but also for prognostication and treatment monitoring (this aspect is insufficiently explored). This is particularly relevant given the aggressive behavior of CCA and the limitations of current imaging and tissue biopsy modalities in tracking disease evolution. Expanding the discussion to include how liquid biopsy could aid in longitudinal disease monitoring, assessment of minimal residual disease, or early detection of recurrence would significantly enhance the clinical relevance of the paper.

Author Response

Reviewer 3

Reviewer Comment 1: The introduction is generally well-structured. However, the final paragraph, which outlines the study objective, requires revision. The sentence could be rephrased for clarity and scientific tone.

Response:
We thank you for this helpful suggestion. In response, we have revised the final paragraph of the Introduction to improve clarity and scientific tone. The updated version now more explicitly states the objective of the review, namely to critically evaluate limitations in conventional diagnostic modalities for cholangiocarcinoma (CCA) and to explore the emerging potential of liquid biopsy combined with CRISPR/Cas-based systems as a next-generation approach for sensitive and specific biomarker detection. This rephrasing aligns more closely with the overall scientific narrative and enhances readability.

Reviewer Comment 2: In the results, I recommend including an initial paragraph summarizing the general characteristics of the included studies, such as study design, sample size, patient population, and methodology, to provide the reader with a broader contextual understanding.

Response:
We appreciate your insightful recommendation. Accordingly, we have added a new introductory paragraph at the beginning of the Results section (Chapter 3.1- Overview). This paragraph now summarizes the general characteristics of the included studies, highlighting aspects such as the time frame of publication (2018–2025), geographic distribution of patient populations (Western and Southeast Asian cohorts), types of methodologies used (liquid biopsy platforms and CRISPR-based assays), and the diagnostic biomarkers analyzed (primarily circulating microRNAs). These contextual details provide readers with a clearer overview of the evidence base supporting our analysis.

Reviewer Comment 3: One area that requires substantially more attention is the molecular profiling of cholangiocarcinoma. Several key biomarkers known to play critical roles in the pathogenesis, diagnosis, and prognostication of CCA are either not mentioned or only poorly discussed. These include TP53, IDH1, IDH2, KRAS, ARID1A, BAP1, BRAF, FGFR2, and HER2. A more comprehensive discussion of these biomarkers and their implications for diagnosis, prognosis, and targeted therapies would strengthen the molecular basis of the review. Additionally, the manuscript should distinguish between intrahepatic and extrahepatic CCA/gallbladder in terms of molecular alterations, as this differentiation is essential for personalized management strategies and reflects the evolving understanding of CCA heterogeneity.

Response:
We fully agree with your comment regarding the importance of incorporating a more detailed discussion of molecular profiling in CCA. In response, we have significantly expanded Chapter 3.2 to include the roles of major genomic alterations such as TP53, IDH1/2, KRAS, ARID1A, BAP1, BRAF, FGFR2, and HER2. We discuss their relevance in the pathogenesis and classification of CCA, along with their implications for prognosis and targeted therapy development. Furthermore, we have clarified the distinct molecular profiles between intrahepatic and extrahepatic CCA subtypes, as well as those associated with liver fluke infections. These additions better reflect the current understanding of CCA heterogeneity and enhance the translational value of the manuscript.

Reviewer Comment 4: While the discussion provides a good overview of future directions and technical challenges, the manuscript falls short in exploring the potential of liquid biopsy not only for diagnosis but also for prognostication and treatment monitoring (this aspect is insufficiently explored). This is particularly relevant given the aggressive behavior of CCA and the limitations of current imaging and tissue biopsy modalities in tracking disease evolution. Expanding the discussion to include how liquid biopsy could aid in longitudinal disease monitoring, assessment of minimal residual disease, or early detection of recurrence would significantly enhance the clinical relevance of the paper.

Response:
We sincerely thank you for this important observation. In response, we have revised the Discussion section to more thoroughly address the broader clinical applications of liquid biopsy in CCA management. The revised text now includes an expanded discussion on how LB could support longitudinal monitoring of disease progression, assessment of treatment response, detection of minimal residual disease, and early identification of recurrence. These additions underscore the clinical utility of LB not only as a diagnostic tool but also as a means of improving real-time disease management in CCA patients, particularly given the limitations of conventional imaging and biopsy modalities.

Round 2

Reviewer 1 Report

Comments and Suggestions for Authors

Dear authors,

Thank you very much for the careful correction of the manuscript. To my opinion ir can be now accepted in its present form.

Kind regards,

Reviewer 2 Report

Comments and Suggestions for Authors

The authors have addressed my comments.

Reviewer 3 Report

Comments and Suggestions for Authors

This is the revision of the manuscript by Sidabraite et al. The authors addressed all of my previous concerns. The study addresses an important topic and is likely to attract significant interest from the journal's audience. I consider the current version suitable for publication.